# Cold Cathodes with Two-Dimensional van der Waals Materials

**DOI:** 10.3390/nano13172437

**Published:** 2023-08-28

**Authors:** Yicong Chen, Jun Chen, Zhibing Li

**Affiliations:** 1State Key Laboratory of Optoelectronic Materials and Technologies, Guangdong Province Key Laboratory of Display Material and Technology, School of Electronics and Information Technologies, Sun Yat-Sen University, Guangzhou 510275, China; 2State Key Laboratory of Optoelectronic Materials and Technologies, Guangdong Province Key Laboratory of Display Material and Technology, School of Science, Sun Yat-Sen University, Shenzhen 518000, China

**Keywords:** cold cathode, 2D van der Waals materials, field emission, coherence, hot electron emission

## Abstract

Two-dimensional van der Waals materials could be used as electron emitters alone or stacked in a heterostructure. Many significant phenomena of two-dimensional van der Waals field emitters have been observed and predicted since the landmark discovery of graphene. Due to the wide variety of heterostructures that integrate an atomic monolayer or multilayers with insulator nanofilms or metallic cathodes by van der Waals force, the diversity of van der Waals materials is large to be chosen from, which are appealing for further investigation. Until now, increasing the efficiency, stability, and uniformity in electron emission of cold cathodes with two-dimensional materials is still of interest in research. Some novel behaviors in electron emission, such as coherence and directionality, have been revealed by the theoretical study down to the atomic scale and could lead to innovative applications. Although intensive emission in the direction normal to two-dimensional emitters has been observed in experiments, the theoretical mechanism is still incomplete. In this paper, we will review some late progresses related to the cold cathodes with two-dimensional van der Waals materials, both in experiments and in the theoretical study, emphasizing the phenomena which are absent in the conventional cold cathodes. The review will cover the fabrication of several kinds of emitter structures for field emission applications, the state of the art of their field emission properties and the existing field emission model. In the end, some perspectives on their future research trend will also be given.

## 1. Introduction 

Compared to traditional thermionic cathodes, cold cathodes have a much higher emission current density with a lower power consumption, faster response time and narrower energy spread. Furthermore, it can be miniaturized via modern micro-fabrication techniques. All these advantages make it promising in the realization of a high-performance vacuum electron source [1,2,3,4,5,6] as well as other novel applications such as flat panel field emission display [7,8,9], parallel electron beam lithography [10,11] and flat panel X-ray source [12,13,14,15,16,17,18].

Since the discovery of graphene [19], the attempts to use two-dimensional (2D) van der Waals (vdW) materials as cold cathodes have attracted much attention due to their following features: (i) covalent bonding, which leads to clean surfaces (less surface states of dangling bonds and surface absorbates) and can be smoothly integrated in various substrates despite lattice mismatching (the absent of dangling bonds and surface absorbates would be helpful for uniform emission; in addition, their electron mobility is usually very large, that is welcome for fast devices); (ii) small local densities of states in the vicinity of the native Fermi level, which enables significant energy barrier lowering due to field penetration in the emission region (high-voltage part of the film or edge) that will favor both surface emission (emission from the 2D surface) and edge emission (emission from the 1D edge); (iii) large surface ratio, due to which the space charge effect of 2D emitters should be smaller than the nanowire emitters, causing a larger saturation current (it would also allow us to control the emission with a magnetic field, not just electric field, as the surface may sustain large magnetic flux); (iv) 1D edge structure, which induces significant field enhancement (although not as large as nanowires of the same height) and offers a wide range of work function modifications; and (v) conserves electrons’ momentum in directions parallel to the surface (in the surface emission mode) or parallel to the edge (in the edge emission mode), and the latter leads to a highly directional knife-line electron beam and carries out phase information of quantum states of the emitter.

So far, people have found that the field emission current density of 2D vdW emitters might not be as excellent as the state of art of quasi-one-dimensional nanowire or nanotube field emitter arrays (especially for flat panel electron sources) [20,21]. However, for applications of point sources or line sources, 2D vdW emitters can be a great candidate because of their monochromaticity and coherence [6,22]. Furthermore, due to their low densities of state, 2D vdW emitters also offer a platform to realize novel vacuum devices based on high-efficiency hot electron emission [23,24], which may overcome the long existing problem of stability for cold cathodes. In theoretical studies, a universal field emission model on 2D vdW materials is still lacking. Even though most of the relevant experiments show the straight F-N plot which is derived from the classical Fowler–Nordheim law [25] (or the improved model of Murphy–Good [26]), their extracted slope and intersection would have complex meanings as pointed out by R.G. Forbes [27]. The theoretical difficulty is mainly due to the field emission of nano-structures basically being a multi-scale, many-body, quantum mechanical problem. Because it is sensitive to the atomic structure of the emitters, it is also difficult to repeat the details of field emission experimental observations. Nevertheless, theoretical studies on ideal models do show that the conventional basic model of 3D metallic emitters (with a 2D emission surface) is not applicable to 2D emitters.

After more than ten years of investigations, we think it is time to make a review on the findings (both in experiments and theory) on 2D vdW field emitters, which are absent in the conventional cold cathodes. Although several recent works have reviewed the field emission properties [28], applications [29] and theory [30] of 2D emitters, some important aspects for cold cathodes such as divergence, coherence and the many-body effect have not been covered, which will be focused here. We hope that this can encourage the people who still engage in this field and guide new researchers.

In this review, we will firstly give a brief introduction on the emitter structure of 2D materials, emphasizing their advantages or disadvantages in cold cathode applications. After that, the state of the art of their unique field emission properties including I-V characteristics, stability, coherent field emission pattern and the existing field emission model will be introduced. Finally, perspectives on the research trend of 2D vdW emitters will also be given.

## 2. Emitter Structure

Due to the geometry structure of 2D materials, 2D emitters can have two distinct structures: edge emitter and surface emitter. In this section, we will give a brief introduction on the structure and fabrication method for each kind of emitter and discuss their advantages and disadvantages for field emission applications. 

### 2.1. Edge Emitter

Generally, the edge structure has a much higher field enhancement factor than that of the plane structure because of the atomic thickness of 2D materials. Therefore, at the early stage, studies on field emission from 2D materials are mainly focused on the edge emitter. Generally, the edge emitter can be divided into vertical and lateral emitter structures, which are defined as the edge structure being vertical to or parallel with the cathode substrate. Details for each structure and the fabrication method will be introduced in the following section.

#### 2.1.1. Vertical Emitter

The simplest and lowest cost method to fabricate large-area vertical edge emitters may be the mechanical exfoliation of the 2D material. Considering that it is difficult to precisely control the morphology of a 2D material film by exfoliation, both the amount and uniformity of edge emitters using this method are usually not good enough for field emission applications. Therefore, only a few works on exfoliated 2D material field emitters have been reported. For example, C. Wu et al. [31] used the adhesive tape to realize a raised edge structure on graphene film. S. R. Suryawanshi et al. [32] used the exfoliation method to realize a black phosphorus (BP) nanosheet field emitter. 

To improve the field emission uniformity of 2D material film, researchers proposed to use several kinds of solution-based methods to realize edge structure. For example, G. Eda et al. [33] used a solution-based spin coating method for deposition of graphene film. By using a relatively low spin coating speed, a raised edge structure can be realized due to the densely distributed graphene sheets over the substrate. Z. Wu et al. [34] fabricated single-layer graphene films with an edge structure by electrophoretic deposition from a stable suspension of graphene. M. Qian et al. [35] used the screen printing method to prepare a graphene cathode. R. V. Kashid et al. [36] used a solution method to prepare a few-layer MoS_2_ with edge structures. C. P. Veeramalai et al. [37] fabricated a few-layer MoS_2_ with edge structures with a hydrothermal method. H. Huang et al. [38] used a solution-based method to fabricate a few-layer Bi_2_Se_3_ with an edge structure. Moreover, C.-K. Huang et al. [39] also used microfabrication techniques to prepare graphene edge emitter arrays. By etching graphene/Cu with a patterned structure, the exposed graphene on the edge of the pattern can fold and become the edge emitter. 

Because the above-mentioned edge emitters usually have a random orientation, it can limit the field enhancement factor and emitter density, which could be a negative factor for field emission application. To realize a high-performance 2D material edge field emitter, the fabrication of a vertical-aligned edge structure is in need. The first investigation of a vertical-aligned 2D material as a field emitter was carried out by A. Malesevic et al. [40], who fabricated vertical-aligned few-layer graphene (FLG) by microwave plasma-enhanced chemical vapor deposition (MPECVD). After that, several groups have also carried out field emission studies of vertical-aligned FLG fabricated by MPECVD [41,42,43,44]. For other vertical-aligned 2D materials, several synthesis methods have been reported. For example, H. Zhong et al. [45] fabricated vertical-aligned SnS_2_ field emitter arrays with a biomolecule-assisted method. H. Li et al. [46] synthesized vertical-aligned MoS_2_ field emitter arrays by using the chemical vapor deposition (CVD) method. P. R. Dusane et al. [47] used a hydrothermal method to prepare vertical-aligned MoSe_2_ field emitter arrays on carbon cloth. M. Kumar et al. [48] used the radio frequency sputter deposition method to fabricate wafer-scale vertical-aligned ReS_2_ field emitter arrays. C. D. Jadhav et al. [49] synthesized vertical-aligned CuSe field emitter arrays with an electrochemical method. 

To make a better comparison, typical scanning electron microscope (SEM) images of the morphology of graphene fabricated by exfoliation, electrophoretic deposition, etching and MPECVD are presented in Figure 1a–d, respectively. It is seen that the vertical-aligned edge structure has a more uniform and dense distribution. Furthermore, vertical-aligned edge structures usually have a thick base and thin tip, which can avoid the swinging of the emitter and is beneficial for heat dissipation. All these features make it competitive for achieving high uniformity and stability in field emissions, which has potential applications in flat panel electron sources. 

#### 2.1.2. Lateral Emitter Structure

Apart from the vertical emitter structure, researchers also realized an edge emitter by using the lateral structure which can be fabricated with a precise and scalable process with a microfabrication technique. For example, H.M. Wang et al. [50] fabricated a graphene nanogap with a few hundred nanometers as shown in Figure 2a. By utilizing the self-collapse of suspended graphene during the drying process, two edges of graphene acting as the cathode and anode can be realized. To avoid the leakage current in the insulator between the nanogap, S. Kumar et al. [51] fabricated a suspended FLG field emission device as shown in Figure 2b. By etching both the FLG and SiO_2_ substrate, suspended FLG edges with a gap down to 50 nm can be fabricated. J.L. Shaw et al. [52] further demonstrated a three-terminal device by using the suspended graphene edges above a gate electrode as the source and drain, and the device structure can be seen in Figure 2c. Because the emission electron transports from one edge to another in this kind of device structure, it can work as a vacuum transistor but is not suitable for field emission electron sources. To realize an electron source using a lateral edge emitter, V. I. Kleshch et al. [53] cut the graphene-covered quartz substrate mechanically. With the exposed graphene on the cleaved edge of quartz, a blade-type electron source as shown in Figure 2d can be obtained. P. Serbun et al. [54] also used the exposed graphene edge to fabricate a point-type emitter by cutting the thin graphene film paper into triangular-shaped pieces. Although a lateral emitter structure can be utilized as a point or line electron source, it seems to be not suitable for flat panel electron sources due to its geometrical feature. 

### 2.2. Surface Emitter

Although edge emitters have a very high field enhancement factor which favors a low turn-on field, they also have a distinct drawback that it is difficult to control each emitter’s structure especially at the atomic scale. According to some references [55,56], the edge structure with different functionalized atoms can largely influence the field emission properties, which may limit the uniformity. To solve this problem, surface emitters including 2D/1D hybrid structures, vdW heterostructures and 2D material suspended structures have been investigated. Each structure and fabrication method will be introduced in the following section.

#### 2.2.1. 2D/1D Hybrid Structure

To increase the field enhancement factor of the surface emitter, researchers proposed to use a substrate with a protuberant structure to support the 2D materials. For example, Z. Yang et al. [57] fabricated a graphene/ZnO nanowire hybrid structure by transferring monolayer graphene to a ZnO nanowire substrate with the usage of polymethyl methacrylate (PMMA) as the supporting layer, and the typical SEM image can be seen in Figure 3a. T. Chang et al. [58] fabricated a graphene/Si tip hybrid structure by transferring graphene to Si tip arrays. D. Ye et al. [59] fabricated graphene oxide (GO)/Ni nanotip arrays by transferring GO onto the Ni nanotip arrays. For other 2D materials, T-H Yang et al. [60] also used a transfer method to fabricate the hybrid structure of MoS_2_ or MoSe_2_ on ZnO nanostructures, and the schematic diagram can be seen in Figure 3b. Apart from the transfer method, X. Shao et al. [6] also fabricated a graphene/Ni tip by using a CVD method to synthesize graphene on the Ni tip. Considering that 2D/1D hybrid emitter arrays have a net-like structure, they are in favor for heat dissipation. High field emission stability was usually obtained from this kind of emitter, which will be introduced in the next section.

#### 2.2.2. Van der Waals Heterostructure

Another method to reduce the applied field for electron emission from a surface emitter is to modify its effective work function, which can be realized using a vdW heterostructure. For example, T. Yamada et al. [61] used a graphene/hBN heterostructure to modify its Fermi level, which helped to obtain a reduced turn-on field. K. Murakami et al. fabricated graphene/hBN/Si [62] and graphene/SiO_2_/Si [63,64,65,66] to realize a metal–insulator–metal (MIM) cathode which can generate hot electrons above the vacuum level of graphene based on internal field emission between the MIM structure. A schematic diagram for the working principle of the MIM cathode can be seen in Figure 4a, where the top graphene layer works as a transparent gate. In another work with the heterostructure of graphene/hBN/graphene as shown in Figure 4b, Y. Chen et al. [24] demonstrated a new type of device with the top graphene layer working as the cathode. Details of its principle will be introduced in Section 4. Compared to the cold cathode based on external field emission, these MIM cathodes based on a vdW heterostructure have a low turn-on voltage which can be suitable for low power applications. Moreover, the stack structure of a MIM cathode can also avoid the contradiction between the emission area and transconductance in traditional gated field emission devices.

#### 2.2.3. Suspended Graphene Structure

Apart from the MIM structure, G. Wu et al. [23] also demonstrated that a suspended graphene structure can have significant electron emission. The schematic diagram of the device principle and its morphology can be seen in Figure 5a,b. By electrically driving the current through the suspended graphene, quasi-equilibrium hot electrons may accumulate in the high-voltage regime. Then, phonons generated by the joule heat may assist electron emission in the direction normal to the surface. Details of the electron emission mechanism will be introduced in Section 4. Because this device utilizes joule heat to drive hot electron emission, the emission efficiency is still overshadowed compared with those of microfabricated field emitters. Furthermore, its response speed may be limited, which may hinder its application in the fields that require a high frequency, and the energy spread of the joule-heat-induced hot electron can also be broad, which makes it not suitable for electron source applications that require high monochromaticity. 

## 3. Field Emission Properties

The field emission measurement of 2D materials can be divided into two parts. One is the measurement of a 2D material film with multi-emitters and the other is the in situ measurement of a single emitter. While the former one mainly concerns the parameters (such as turn-on field and field enhancement factor β) that are related to the performance as the electron source, the latter one focuses on the electron emission mechanism. Their details will be introduced below.

### 3.1. Film Measurement

For the field emission measurement of 2D material films, a typical diode structure consisting of a metal plate or transparent ITO glass as the anode and the 2D material film as the cathode is usually used, and the schematic diagram can be seen in Figure 6a. Early studies on the field emission properties of 2D material films can be dated back to 2008 [33,40], where the turn-on field for graphene was found to be as low as 1 V/μm. After that, the field emission from other 2D material films, such as MoS_2_ [36], MoSe_2_ [67], WS_2_ [68], SnS_2_ [45], SnSe [69], Bi_2_Se_3_ [38], ReS_2_ [48] and CuSe [49], has also been studied. It is worth noting that the turn-on field here is defined as the field for obtaining 10 μA/cm^2^ unless otherwise mentioned. To improve the performance of field emitters, investigations on the effect of morphology, conductivity, surface work function and back contact resistance on their field emission properties are usually performed in the history of nano-cold cathode development [70]. Two-dimensional material field emitters are also no exception. For example, U. A. Palnitkar et al. [71] investigated field emission properties from doped and undoped graphene fabricated by the arc discharge method. They found that the N-doped graphene has the lowest turn-on field of 0.6 V/μm due to its higher Fermi level. N. Soin et al. [43] also performed field emission measurements on the N-doped FLG fabricated using in situ N_2_ plasma treatment. Due to the work function reduction, conductivity increasement and microstructure change, its turn-on field can be reduced from 1.94 to 1 V/μm, which typical result can be seen in Figure 6b. Y. Zhang et al. [41] manipulated the morphology of FLG by adjusting the growth time and gas ratio during the MPECVD process. They found that the optimal shape had fewer layers, sharp corners, large height and was free of amorphous carbon, which can carry a maximum current density of 7 mA/cm^2^. Moreover, they also investigated the effect of the substrate and found that the interfacial contact resistance of FLG and the substrate play an important role in the field emission properties [44]. By using a stainless-steel substrate, a larger emission current of 35 μA at the field of 160 V/μm with an estimated area of 0.00449 μm^2^ can be obtained compared to other substrates of silicon, quartz and carbon cloth. To further increase the field enhancement factor β, J.-h. Deng et al. [72] fabricated FLG on CNT by using a radio frequency sputtering deposition method. By utilizing the high aspect ratio of CNT as the substrate, a β of ~4398 can be obtained from the FLG on the CNT sample, which leads to the lowest turn-on field of 0.956 V/μm among their samples. E. Stratakis et al. [73] fabricated FLG on Si microspikes using a solution-based method. Due to its high value of β, a low turn-on field of 2.3 V/μm can be obtained, which can be seen in Figure 6c. T.-H. Yang et al. [60] investigated the effect of morphology on the field emission properties from a hybrid field emitter of MoS_2_ or MoSe_2_ supported on different ZnO nanostructures. By using ZnO nanotapers as the substrate, a low turn-on field of 7 V/μm can be obtained from the MoSe_2_/ZnO nanotaper hybrid emitter due to the sharpness of the ZnO nanostructure, which can be seen in Figure 6d. 

To analyze the field emission I-V curve, a Fowler–Nordheim (F-N) plot of ln(I/V^2^) versus 1/V is usually carried out, where the model is based on the metal cathode with a flat surface. Although the 2D electron gas in 2D materials should lead to a modified F-N plot in the form of ln(I/V^α^) versus 1/V^β^ [74], most of the related works still plotted their results in the classical way and a linear plot was usually obtained. Nonlinear F-N plots were only reported in a few works and the underlying mechanism can be attributed to current saturation, joule heat, surface state emission and electron confinement, which are similar with the case of 1D emitter. For example, in the F-N plot of graphene on ZnO nanowire as shown in Figure 7a, two sections with upward bending in the high field region were observed, which might be caused by the confinement of electrons in a 2D system [57]. A similar feature of the F-N plot has also been reported in SnSe as shown in Figure 7b, which was believed to be related to a different emission site from the SnSe film [69]. In the F-N plot of WS_2_ as shown in Figure 7c, downward bending in the high field region due to current saturation was reported [68]. In the modified F-N plot of the MoSe_2_/ZnO nanotaper and MoS_2_/ZnO nanotaper hybrid emitter as shown in Figure 7d, three regions can be identified due to the conduction band current saturation and valance band electron emission in the N-type semiconductor [60]. 

Apart from the field emission I-V characteristics, the field emission stability of 2D material films is also important in cold cathode applications. A comparison of the stability among several typical 2D material field emitters is shown in Figure 8, where Figure 8a–e are the results for screen-printed graphene [35], exposed graphene [39], vertical-aligned FLG [41], surface emitter of graphene/ZnO nanowire [57] and a transition-metal dichalcogenide (TMD)/ZnO nanotaper hybrid structure [60], respectively. Compared to the screen-printed graphene (fluctuation of ~10%) and exposed graphene (fluctuation of ~10%), the vertical-aligned FLG with a fluctuation of ~3.7% and surface emitter of 2D/1D hybrid structure with a fluctuation of ~5% show higher stability, which is mainly due to its secure structure that is hard to swing during field emission. Moreover, the field emission uniformity is also an important factor for the application of flat panel electron sources, which has been measured in a few related works. Figure 9 presents the field emission pattern of several 2D material films where (a), (b) and (c) are the results for exfoliated BP [32], screen-printed graphene [35] and vertical-aligned FLG [41], respectively. It is seen that the screen-printed graphene and vertical-aligned FLG have a more uniform emission site. However, they are poorer than the state of the art of quasi-one-dimensional nano-field emitters [21], which makes them not so competitive in flat panel electron sources. To further improve the uniformity, one needs to fabricate the 2D emitter with not only a uniform spatial distribution but also a uniform surface work function as well as a uniform resistance. Using a surface emitter such as a 2D/1D hybrid emitter may be a solution due to its net-like structure. But related results have not been reported yet. More results of field emission properties from 2D material films have been listed in Table 1. It is clearly seen that the turn-on field for vertical edge emitters is much smaller than that for surface emitters, which is attributed to their larger field enhancement factor. This indicates a higher emission efficiency for the vertical emitter as mentioned before.

### 3.2. In Situ Measurement 

To exclude the average effect of multi-emission sites from 2D material films, in situ field emission measurement has been carried out under the SEM/transmission electron microscope (TEM) chamber by using a nano-manipulated metal tip as the anode, which is the same as the works of other nano-cold cathodes. The first investigation on the local field emission characteristics of single-layer graphene was performed by Z. Xiao et al. [74] in 2010. They measured field emission from the edge of graphene in a SEM chamber and explained their field emission properties using a modified F-N theory. Plotting the result in the form of ln(I/E^α^) versus 1/E^β^, it is found that (α, β) = (3/2, 1) under the high field regime and (α, β) = (3, 2) under the low field regime, which can be seen in Figure 10a,b. After that, several works on the in situ field emission measurement of graphene or graphene-related thin film have been performed. For example, J. Xu et al. [81] measured field emission from the surface position of graphene suspended by two electrodes. They found a transition from space charge flow at low bias to the F-N theory at a high current emission regime, which can be seen in Figure 10c. S. Tang et al. [82] observed the joule heating effect from the field emission of single FLG in a TEM chamber, which typical upward bending I-V curve in the high field region can be seen in Figure 10d. To avoid the joule-heating-induced vacuum breakdown, they proposed to use the graphite interlayer between FLG and the tungsten substrate.

Similar results have also been reported in the studies of other 2D materials. For example, F. Urban et al. [83] investigated field emission characteristics from the surface position of MoS_2_ bilayers in a SEM system. They found that their results can be well described by a modified F-N plot with ln(I) versus 1/E, which can be seen in Figure 11a. A. Pelella et al. [84] fabricated a back-gate-controlled field emission device based on MoS_2_ as shown in the inset of Figure 11b. By increasing the back gate voltage, the electron affinity of MoS_2_ can be lowered, which resulted in a larger current under the same anode voltage provided by the anode probe above the device. Y. Chen et al. [85] performed in situ field emission measurements on an individual hybrid emitter of WSe_2_ on ZnO nanowire in a nanoprobe system. They found that a “tip contact” structure as shown in the inset of Figure 11c is in favor for lower turn-on field and higher stability due to the hotter injecting electron. In an “edge-contact” structure as shown in Figure 11d, the hot electron needs to transport through a suspended WSe_2_ region, while in a “tip-contact” structure, the hot electron can directly transport through the thickness of WSe_2_. A shorter transportation length leads to a higher effective electron temperature, which can lower the effective barrier height. Moreover, the suspended P-type WSe_2_ caused a depletion region during field emission, which resulted in current saturation in the F-N plot as shown in Figure 11d, consistent with the mechanism. 

Apart from the field emission I-V characteristics, field emission microscopy measurements on the edge of reduced graphene oxide (RGO) were also studied by H. Yamaguchi et al. [22], where the fringe pattern was observed as shown in Figure 12i–iii. By adjusting the emission spots of ~1 nm in diameter separated by ~2 nm with coherence, the interference pattern shown as Figure 12v can be simulated, which is similar to the experimental result as shown in Figure 12iv. Other results of in situ field emission properties from 2D materials have been listed in Table 2. It is seen that different from the film measurements, most of the results from in situ measurements do not follow the classical F-N law, which reflects the intrinsic field emission properties of 2D emitters. 

## 4. Theoretical Model

Electrons of 2D vdW materials can move freely in an atomic layer but are confined tightly in the direction normal to the layer surface. Therefore, electron emission in the normal and paralleled directions of 2D materials are different, which should lead to distinct field emission properties. To have a better understanding of this difference, studies on 2D material field emission theory can be divided into two parts. One is for the edge emission mode and the other is for the surface emission mode, which will be introduced below. 

### 4.1. Edge Emission Mode

The most significant feature of the edge emission mode is that the electron emits from a line with atomic thickness. If the system (including the applied field) has translational symmetry along the edge, the emission wave will preserve the lattice–wave–vector component along the edge direction. It is known that the edge electronic structure and the vacuum barrier strongly depend on the edge type and how the edge is saturated. In the following, we will review theoretical studies about metallic nanowall model and graphene edge emission.

#### 4.1.1. Classical Nanowall Model

A simple model for the 2D emitter in edge emission mode is the nanowall model, in which electrons of the emitter are treated as free-electron gas in equilibrium or quasi-equilibrium. The nanowall model does not contain information of atomic orbitals or the electron band structure of the emitter, except the Fermi level and work function. 

Particularly, a nanowall is mounted on a flat cathode plane perpendicularly, as presented in Figure 13, where the anode plane is parallel to and far away from the cathode plane. The translational symmetry along the edge allows us to obtain the exact electrostatic potential in the vacuum region by a powerful mathematical method called conformal transformation. The local electrostatic field on the surface of the nanowall has been obtained by R. Miller et al. [92], from which they derived the explicit expression for the enhancement factor at the middle line of the edge as
(1)$$\gamma_{S}=\sqrt{\frac{\pi }{2}\frac{h}{w}}$$
and electrostatic field in the vicinity of either corner of the wall as
(2)$$F(z,x)=\gamma_{S}F_{M}{\left(\frac{2w}{3\pi \sqrt{{\left(z-w/2\right)}^{2}+x^{2}}}\right)}^{1/3},\;\mathrm{for}\;(z,\;x)\to (w/2,\;0).$$

On the other hand, due to the confinement of the side surfaces, electron densities of quantum states which are stationary waves in the width direction vanish at the corners, hence the emission exactly from the corners is also vanishing. The elementary equations for cold field emission from nanowalls including the sideband effect have been given by X.-Z. Qin et al. [93]. In their model, each electron of the emitter occupies a different quantum state, denoted by *Q*, which would emit to the vacuum independently from the edge. For simplicity, the local barrier field can be described by a uniform field *F* (it would be interesting to explore the position-dependent *F* for an atomically thin nanowall). The magnitude of *F* is related to the applied field *F_M_* by the field enhancement factor *γ_S_* as *F* = *γ_S_F_M_*. An electron in the state *Q* of energy *W_Q_* (relative to the potential well base of the emitter) contributes a sub-current density, which may be written in a general form
(3)$$j_{Q}=e\Pi_{Q}f(W_{Q},T)D_{Q}(F,H_{Q}).$$

Here, *e*Π*_Q_* is the electron–current–density component (for state *Q*) approaching the emitting surface in a direction normal to it, and *f*(*W_Q_, T*), known as the Fermi–Dirac distribution function, is the occupancy of state *Q* at the temperature *T*. The transmission coefficient *D_Q_* is a function of *F* and *H_Q_* (the barrier field at the edge and the barrier height seen by an approaching electron in state *Q* when *F* = 0).

In the free-electron gas model, the energy of the electron confined in the nanowall is given by $W_{Q}=W_{x}(p_{x})+W_{y}(p_{y})+W_{z,n}$, where $W_{x}=p_{x}^{2}/2m_{e}$ is the forwards energy, $W_{y}=p_{y}^{2}/2m_{e}$ is the lateral energy and $W_{z,n}=n^{2}W_{z,1}$ is the sideband-confined energy with $W_{z,1}=\pi^{2}\hbar^{2}/2m_{e}w^{2}$. The positive integer n labels the sidebands. 

By treating the nanowall as a wedge, the image potential energy is given by
(4)$$U_{image}(a,\gamma )=-\frac{e^{2}}{{\left(4\pi \right)}^{2}\varepsilon_{0}}\frac{1}{a}\left(1+\frac{\pi -\gamma }{\sin\gamma }\right),$$
where *a* is the distance from the edge line to the electron and *γ* is the angle of the radius vector with respect to one of the wedge planes. 

In the Jeffreys–Wentzel–Kramers–Brillouin (JWKB) approximation, the emission current line density from the n-th sub-band can be calculated as
(5)$$J_{n}=\frac{e\sqrt{2\pi m_{e}}}{2\pi^{2}\hbar^{2}}\left\{\frac{\pi k_{B}T/d_{n}}{\sin(\pi k_{B}T/d_{n})}\right\}d_{n}^{3/2}\exp(-G_{n}),$$
where $G_{n}=g_{e}\int_{x_{n0}}^{x_{n}}\sqrt{H_{n}-eV(x,F_{M})+U_{image}(x)}dx$ is the JWKB exponent and $d_{n}={\left(\partial G_{n}/\partial H_{n}\right)}^{-1}$ is the decay width. The total emission current line density is $J=\sum_{n=1}^{\infty }J_{n}$. 

One can see that the F-N plot of the model is generally not a straight line; that has been somehow confirmed by a number of experiments [74,81] which can be seen in Figure 10.

#### 4.1.2. Graphene Edge Emission

When the thickness of a 2D emitter is comparable to the atom spacing, the previous classical nanowall model is inadequate. The dispersion relation and the edge atomic structure become crucial, and the field penetration should be considered. Graphene is a representative vdW material that is a monolayer of carbon atoms. The edge emission mode of graphene with a uniform edge has two remarkable features which are commonly shared by 2D emitters. First, it preserves the translational symmetry along the edge when the applied electric field is normal to the edge and parallel to the plane of graphene. Due to this symmetry, the lattice momentum in the direction parallel to the edge is conserved and encoded in the emission orientation. Second, the tunneling barrier has a minimum ridge in front of the edge and enlarges the size of the surface atomic orbital image viewed from the vacuum side of the barrier. It leads to the self-focusing effect of a 2D emitter: the emission dispersed angle related to the emitter plane is inversely proportional to the size of the orbital image instead of to the size of the orbital itself if the emitter is a chain of atoms. Therefore, the emission may have good momentum resolution along the edge direction and good spatial resolution in the direction vertical to the emitter plane.

We will introduce two theoretical approaches which incorporate the atomic lattice structure and electronic structure of 2D emitters, using a graphene emitter as a model. The first one concentrates on the emission current and geometric optic effect. The second one retains phase information of the quantum states of an emitter.

Before the introduction, some points on the properties of graphene need to be clarified. First, graphene has a linear dispersion relation $E(k)=\pm \hbar v_{F}|k|$ according to the single-band tight binding model. Taking the degrees of spin and valley into account, the density of states (per unit area) may be given by
(6)$$D(E)=\frac{2}{\pi {\left(\hbar v_{F}\right)}^{2}}|E|.$$

Second, the zigzag edge of graphene has an effective zero-field barrier height *W_eff_* which is higher than the work function *W*_0_ by *t*~2.8 eV as electrons in the vicinity of the Fermi level have to spend the hopping energy t to move in the direction parallel to the zigzag edge. In contrast, the armchair edge emission of the states in the vicinity of the Fermi level may use all the kinetic energy for the motion towards the armchair edge, hence the lowest barrier height is just *W*_0_ [94]. Third, the field penetration at the armchair edge is significant because it has no edge states and the bulk density of states vanishes at the native Fermi level. Supposing that graphene is vertically mounted on a metallic infinite plane, the distance between the armchair edge and the metallic plane is *h*. Under the field of *F*, the penetration potential energy has been estimated as [95]
(7)$$V(x)=-\hbar v_{F}\sqrt{\frac{2\pi \varepsilon_{0}F(x+h)}{e\sqrt{-x(x+2h)}}}.$$

(1)Squeezed beamwidth

If emission is from a 3D emitter, an electron wave emitting from a surface atomic orbital of size *ξ_p_* is similar as a wave comes out from a hole of the same size. The emitting electron wave will immediately obtain a transverse momentum of $\hbar /\xi_{p}$ when leaving the vacuum barrier according to the uncertainty principle. On the other hand, the emitting electron at the outer classical turning point has zero normal momentum. The electron will be accelerated by a uniform electric field normal to the anode surface to obtain a forward momentum $\hbar k_{a}$ at the anode plane. Therefore, the divergence angle of such emission wave is 2/(*k_a_ξ_p_*), which divergence angle is defined as the angle of two asymptotic profile lines of the beam in the z-x plane. 

For a 2D emitter, a universal effect of its vacuum barrier is that the beamwidth in the normal direction of the emitter plane is squeezed [96]. The isopotential surface in the vicinity of the edge of a 2D emitter has large curvature as shown in Figure 14, which amplifies the size of the edge atomic orbital like a convex lens. The effective size of the orbital in the conformal transformed space, viewed from the vacuum side of the barrier, is ${\bar{\xi }}_{p}=\sqrt{2\xi_{p}h}$, which could be much larger than *ξ_p_*, the real size of the orbital, for a 2D emitter with a large height. Because the accelerating field in the conformal transformed space $\left(\tilde{x},\tilde{z}\right)$ is along the $\tilde{x}$ axis and uniform, the beamwidth spreading as a plane wave goes through a hole of size ${\bar{\xi }}_{p}$. So, the divergence angle in the $\tilde{z}$ direction is $2/(k_{a}{\bar{\xi }}_{p})$. The forward momentum is the same as from the 3D emitter if it is measured on a screen with a distance away from the emitter. Hence, compared to the divergence angle of the 3D emitter, the previous discussed effect results in a reduction factor of $\sqrt{\xi_{p}/2h}$ for the 2D emitter. 

For an atomic orbital of binding energy *W_p_*~*W_eff_*, the kinetic energy is of the same order. The orbital size may be estimated by Heisenberg’s uncertainty principle, which is $\xi_{p}=\hbar /\sqrt{2m_{e}W_{p}}$. If *W_p_* = 4 eV and *h* = 2 μm, the reduction factor will be 0.005. If the anode plane with a voltage of 10^6^ V is 1 cm away from the cathode plane, the beamwidth at the anode will be 100 nm. 

(2)Emission wave and interference pattern

According to quantum mechanics, electrons have wave behavior, and that is the fundamental reason for the field emission tunneling. In principle, coherent interference of the emitting electron wave contains information of the geometric structure, the electron energy band (a long wavelength property), the local atomic orbitals (a short wavelength property) and dynamic properties of the emitter. Excellent coherence is inherent in cold field emission (CFE) for two reasons: the emission energy is restricted within a narrow range about the Fermi level by the tunneling barrier and thermal fluctuation is almost irrelevant for the tunneling process. The quantum mechanical confining effect may further enhance the coherence of CFE from a nano-emitter.

To describe the coherence emission, one needs to go beyond the Fowler–Nordheim theory. The difficulty is how to connect the electron wave in the emitter to the electron wave propagating in the vacuum. The path-decomposed Green’s function method (PDGFM) has been developed in Ref. [95] that decomposes the emission path into a path inside the emitter and a restricted path in the vacuum. A separating surface Ω is shown with a yellow dashed line in Figure 15, where the left side of Ω is the atomic potential dominated region (APDR) and the right side of Ω is the vacuum potential region (VPR). An electron wave propagating inside the emitter is described by a Green’s function of the emitter Hamiltonian. The propagation in the vacuum is described by a restricted Green’s function that can be obtained by solving the Schroedinger equation of a single electron in the vacuum. 

The PDGFM was first demonstrated with graphene in the edge emission mode where the field-emitting electrons are mainly from localized atomic orbitals located in the vicinity of the emission surface. It can be generalized to nano-emitters of vdW materials that are described, for instance, by the tight-binding Hamiltonian $H_{0}=\sum_{jj^{\prime },\alpha \alpha^{\prime }}t_{jj^{\prime }}^{\alpha \alpha^{\prime }}a_{j}^{(\alpha )+}a_{j^{\prime }}^{\left(\alpha^{\prime }\right)}$. The wave of frequency ω emitting from the position of ***r″*** for large times is strictly given by the path decomposition formula
(8)$$\psi_{\alpha }^{em}(r,\omega )=\int_{\Omega }G^{r}(r,r^{\prime })\frac{i\hbar }{2m}{\overset{↔}{\partial }}_{n}\int_{r^{\prime \prime }}G(r^{\prime },r^{\prime \prime })\psi_{\alpha }(r^{\prime \prime }),$$

The second integral is over ***r″***, the position of the electron in the emitter. *ψ_α_* is the α-th eigenfunction of *H*_0_ and *G*(***r′***, ***r″***) is the retarded Green’s function of *H*_0_. The first integral is over ***r′*** on the separating surface Ω. The two-direction derivative is defined as ${\overset{⃡}{\partial }}_{n}={\overset{\leftarrow }{\partial }}_{n}-{\vec{\partial }}_{n}$, and G^r^(***r′***,***r″***) is the restricted retarded Green’s function of a single propagating electron in the VPR with the vacuum potential. The restricted Green’s function *G^r^*(***r′***, ***r″***) is given formally by the path integral
(9)$$G^{r}(r^{\prime },r^{\prime \prime })=\sum_{\Gamma }e^{-\frac{i}{\hbar }S[\Gamma ]}.$$

The summation is over the restricted paths (Γ) which do not touch Ω except the starting point ***r′***. The action *S*[Γ] may be calculated in the JWKB approximation [95]. 

To show the power of PDGFM, coherent field emission of the zigzag edge of graphene with and without magnetic field are predicted as shown in Figure 16, where (a) is the dragonfly-like field emission pattern without a magnetic field [95] and (b) is the emission pattern under a magnetic field of 15 T [97]. Two interference fringes in Figure 16b originate from the wave function of the π-orbital, which manifests the structure of Landau levels. The peaks at larger *y* are due to the magnetic field that breaks the time y parity.

Incorporating the optical excitation, phonon scattering and thermal relaxation, the field emission pattern is also predicted by M. Luo et al. [96] by solving the steady Boltzmann equation. The continuous laser pumps electrons to certain Landau levels and the phonon scattering leads to thermal relaxation. When the optical excitation and thermal relaxation are balanced, a steady distribution is formed. The levels matching the laser frequency have larger occupation probability. On the other hand, the phonon scattering broadens the occupation probabilities. Therefore, the laser is more significant for emitters having discretized levels, such as graphene under a strong vertical magnetic field. In principle, the emission pattern, which sensitively depends on the electron distribution in the edge states, may be manipulated by the laser. 

The coherent emission patterns of graphene in the absence of a magnetic field were observed by Yamaguchi et al. [22], which can be seen in Figure 12. Under the applied voltages of 2.4 kV, they observed a fringe pattern that is dark in the forward direction flanked by two bright emission cones. It would be evidence of the edge π state, as the theoretic prediction of Figure 16a. However, most theoretical predictions for the coherent emission have not been verified by experiments so far. One obvious reason is that the coherent emission is very sensitive to the atomic structure of the emitters [98]. The experimental set-up may also change the emission pattern dramatically. The ab initio study is possible [55,99], but the calculations are limited in small systems under ideal conditions, which make it only suitable for sophistical interests. Up until today, quantitative comparisons between theoretical and experimental results about the coherence emission of graphene are lacking. As far as we know, experiments of graphene emission under a strong magnetic field have not been carried out, neither for the optical excited field emission.

### 4.2. Surface Emission Mode

Besides field emission from edges of 2D emitters, significant surface emission has been observed in experiments as introduced above. Due to the lack of forward energy in the vertical direction of 2D materials, direct surface emission should be extremely weak as calculated by B. Lepetit [100]. The direct surface emission model as well as several existing models involving hot electron injection or thermal emission will be introduced below. 

#### 4.2.1. Direct Surface Emission

When an electrostatic field is applied normal to the surface of a 2D emitter, field electron emission could happen in principle. However, it is extremely difficult to extract an observable emission by an applied field because electrons in the 2D material have no motive energy in the normal direction and the field enhancement is absent. For instance, the minimum barrier height for the surface emission of graphene is as high as 12.81 eV. Anyhow, the theoretical study does reveal some interesting features of direct surface emission which would be useful for some applications.

B. Lepetit [100] found that the most probable direct surface emission is not from the states in the vicinity of the Fermi level but instead from six states with small parallel momenta. Due to the translational symmetry of 2D surfaces, the parallel momenta are conserved in the direct surface emission. He gave the direct surface emission current with the Bardeen transfer Hamiltonian formalism as
(10)$$I\left(F\right)=2\sum_{k}I_{k}\left(F\right),\;I_{k}\left(F\right)=q\frac{2\pi }{\hbar }{\left|\iint_{x,y\in s}dxdyM_{k}\left(r,z_{0};F\right)\right|}^{2}.$$

The emission wave *Ψ**_k_***(***r***,z_0_;*F*) is connected to the material valence electron orbital *Φ**_k_***(***r***,z_0_) at a surface position z_0_ through the matrix element
(11)$$M_{k}\left(r,z_{0};F\right)=\frac{\hbar^{2}}{2m}\left(\Psi_{k}{\left(r,z_{0};F\right)}^{*}\frac{d\Phi_{k}\left(r,z_{0}\right)}{dz}-\Phi_{k}\left(r,z_{0}\right)\frac{d\Psi_{k}{\left(r,z_{0};F\right)}^{*}}{dz}\right).$$

Approximated analytic expression as well as numerical results were obtained with the tight-binding Hamiltonian. As the direct emission current is generally very small, he argued that the significant emitting levels observed experimentally should originate from defects (such as ripples, contaminations, edges, etc.).

#### 4.2.2. Phonon-Assisted Surface Emission

In the in situ experiment of X. Wei et al. [101], the surface emission current from an individual graphene nanoribbon could be collected by the anode probe when a voltage difference of several volts was applied to the two ends of the graphene nanoribbon. The authors argued that electrons are emitting from the surface rather than the edge of the graphene ribbon via the mechanism of phonon-assisted electron emission. In this model, electrons are pumped into the graphene through applying the voltage difference on the two ends of the graphene ribbon. The injecting electrons are accelerated by the internal electric field. In the process, they accumulate kinetic energy and form hot electron gas on the high-voltage side of the ribbon, through scattering with phonons. The effective temperature of the hot electron gas could be high enough for significant thermal emission, as they estimated by solving the Boltzmann equation for the electron distribution in a similar work about electron emission from carbon nanotubes [102]. Though the final step of electron emission is thermionic, the temperature of the graphene emitter in this experiment is estimated to be less than 1500K, and that may be barely named as a new kind of cold cathode. It would provide a current density as large as 12.7 A/cm^2^, but the efficiency (emission current over the pump current) is about 0.1%, which would be too low for applications.

As an effort for increasing the emission efficiency, they proposed a tunneling electron-emitting diode structure [103], in which electrons are injected into graphene and some of the hot electrons are scattered into the vacuum at the high-voltage graphene–anode interface.

#### 4.2.3. Auger Effect in Metal–Insulator–Graphene Heterostructure

Another kind of vertical emission of hot electron gas from graphene was reported by Y. Chen et al. [24] in the vacuum electron emission of a graphene/hBN/graphene heterostructure. Generally, in a MIM-type planer cathode, a field-injecting hot electron can only emit into a vacuum when the driving voltage is in the forward direction and the driving potential energy is larger than the surface work function of the top electrode. However, in their work, abnormal electron emission was observed when the driving potential energy was smaller than the surface work function of graphene and even when the driving voltage was in the reversed direction. They proposed a hot-hole-induced Auger electron emission model to explain the results. The basic idea of the model is that Auger recombination between the field-injecting hot hole and the native cold electron can occur in graphene. When the driving voltage is in the reversed direction, the hot-hole-induced Auger process occurs in the top graphene layer as shown in Figure 17a. Auger electrons with energy higher than the surface barrier can emit into a vacuum, forming the vacuum emission current. While the driving voltage is in the forward direction, the Auger process will occur in the bottom graphene as shown in Figure 17b. Because the Auger electron has a higher energy level than its Fermi level, it can tunnel through the heterostructure and emit into a vacuum under a smaller driving potential energy. By substituting the Auger electron energy spectrum into the F-N tunneling formula, their results in both directions of driving voltage can be well fitted. To improve its emission efficiency (the vacuum emission current over the tunneling current within the heterostructure), they further investigated the influence of the hBN thickness on the efficiency and found out the optimal thickness is ~11 nm [104]. Although the efficiency of this kind of electron emission is still low (~1%), it provides a way to lower the turn-on voltage for MIM planar cathodes. Moreover, it also demonstrates a hot hole vacuum device.

## 5. Perspective

Based on the above comprehensive review, it is seen that a vertical edge emitter has a high emission efficiency while a surface emitter has a high stability. Depending on the requirement of applications, one may choose a different type of emitter as the cathode. Further optimization on their field emission properties may also be performed by using a post-treatment method such as the laser structuring method [105,106]. Apart from the comparison on two types of emitter structures, some perspectives for the future studies of 2D vdW emitters are given as follows:Two-dimensional vdW materials enable cold field emitters to have more flexible structures. A close theoretical description for edge emission is possible, but the present experiments have not reached the precision to test details of the edge emission of 2D emitters, such as the angle dependency of the emission current and coherent emission patterns. Both experiments and theoretical studies find that the edge emission of 2D emitters is significantly weaker than the emission of nanotips of the same height. One simple reason is that the field enhancement factor is proportional to the square root of height/width as Equation (1), while it is of the order of height/width for 1D emitters. Considering that 2D electron systems have rich interesting physics (such as Landau levels, quantum hall effect, spin wave and valley polarization), they would be useful in the weak current applications, such as quantum state read-out and coherent single-electron sources.The many-body effect in 2D vdW materials needs to be further investigated because it provides an important platform for hot carrier vacuum devices. The phonon-assisted surface emission involves a complex mechanism and a precise description is still lacking. In principle, it could provide a large free-electron current. But the Joule-heat associated with the large driven current would be a bottleneck for applications. On the other hand, studies on stacked 2D cold emitters have just started. Many possible options of 2D vdW materials and their combinations would lead to novel emission properties.The field electron emission 2D vdW materials in either edge or surface mode are sensitive to the atomic structure of the edge/surface. Designable defects and edge/surface decorating would change the emission properties dramatically.The multi-field control of electron emission should have broad applications. The 2D emitters have a large surface–body ratio so they may be easier to be controlled by magnetic field or laser beam, in comparison with the solid nanotips that have a small area to receive the fields. One may think about a flexible 2D emitter that could respond to mechanical deformation.

## Figures and Tables

**Figure 1 nanomaterials-13-02437-f001:**
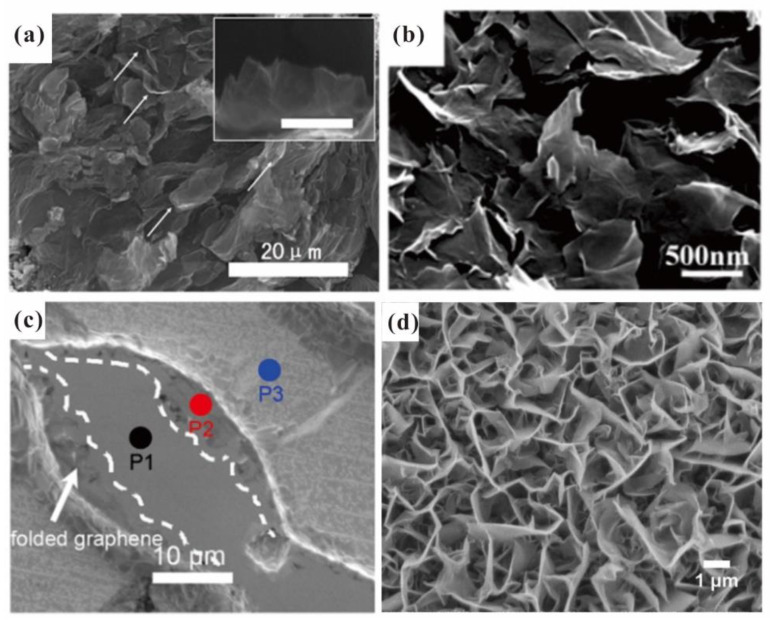
Morphology of graphene vertical edge emitter arrays fabricated by (**a**) exfoliation, (**b**) electrophoretic deposition, (**c**) patterned etching and (**d**) MPECVD. (**a**) Reproduced from [31], with the permission of Elsevier, 2013. (**b**) Reproduced from [34], with the permission of John Wiley and Sons, 2009. (**c**) Reproduced from [39], with the permission of AIP publishing, 2011. (**d**) Reproduced from [40], with the permission of AIP publishing, 2008.

**Figure 2 nanomaterials-13-02437-f002:**
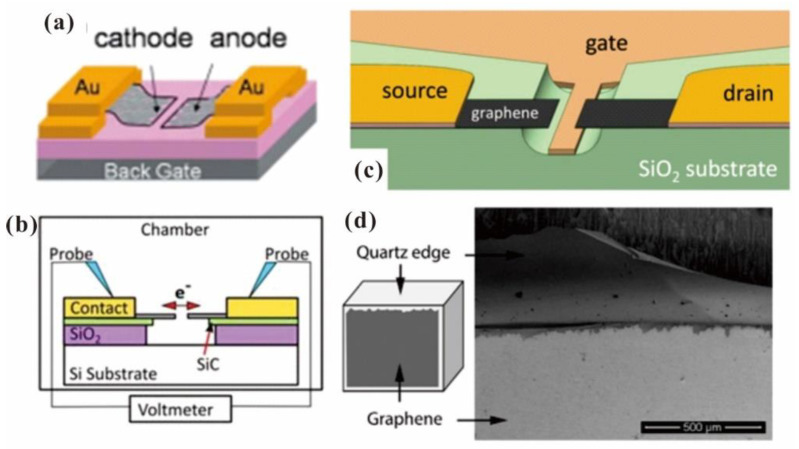
Schematic diagram of several lateral edge emitters. (**a**) Graphene nanogap diode. Reproduced from [50], with the permission of AIP publishing, 2010. (**b**) Suspended graphene nanogap diode. Reproduced from [51], with the permission of AIP publishing, 2014. (**c**) Suspended graphene nanogap triode. Reproduced from [52], with the permission of AIP publishing, 2019. (**d**) Graphene blade-type emitter. Reproduced from [53], with the permission of Elsevier, 2015.

**Figure 3 nanomaterials-13-02437-f003:**
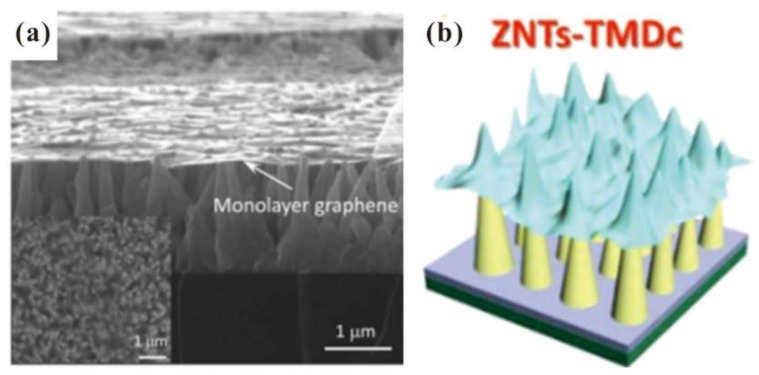
Morphology and schematic diagram of 2D/1D hybrid structure. (**a**) SEM image of monolayer graphene/ZnO nanowire. Reproduced from [57], with the permission of AIP publishing, 2012. (**b**) Schematic diagram of 2D material on ZnO nanostructures. Reproduced from [60], with the permission of John Wiley and Sons, 2018.

**Figure 4 nanomaterials-13-02437-f004:**
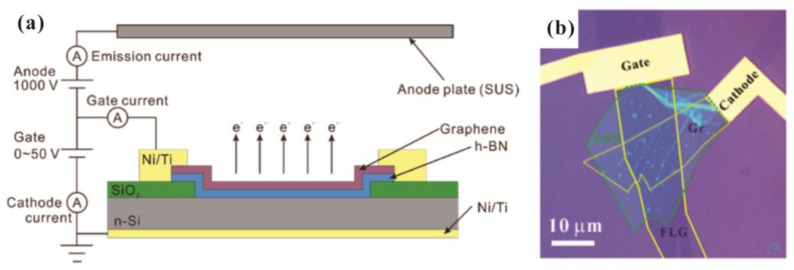
(**a**) Schematic diagram and (**b**) optical image of a graphene-based MIM cathode. (**a**) Reproduced from [62], with the permission of ACS publications, 2020. (**b**) Reproduced from [24], with the permission of ACS publications, 2020.

**Figure 5 nanomaterials-13-02437-f005:**
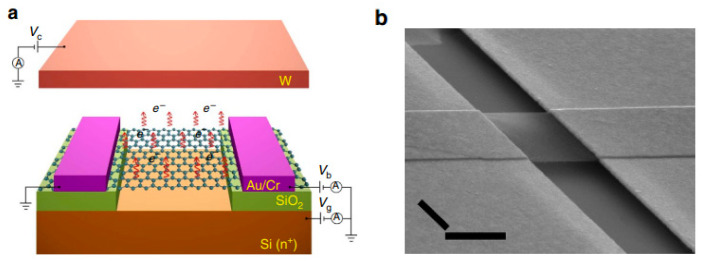
(**a**) Device principle and (**b**) morphology of a suspended graphene emitter (scale bar, 1 μm). Reproduced from [23], with the permission of Springer Nature, 2016.

**Figure 6 nanomaterials-13-02437-f006:**
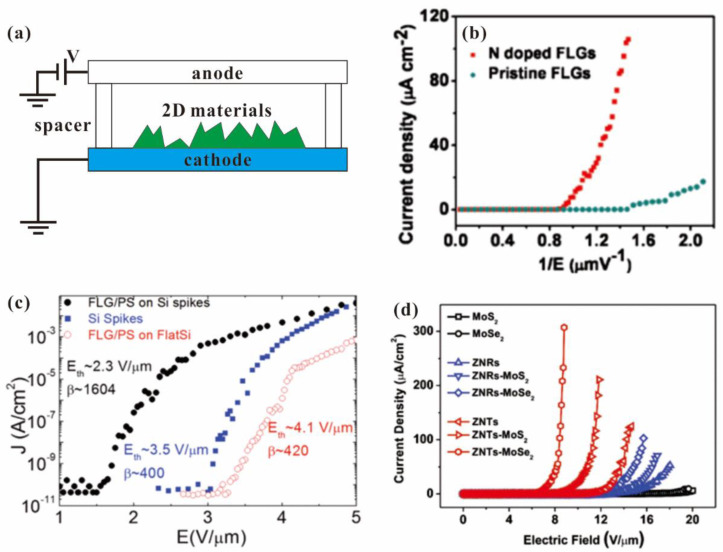
Field emission measurement setup and result of 2D material films. (**a**) Schematic diagram of the setup. (**b**) Dependence of doping on field emission properties of graphene. Reproduced from [43], with the permission of ACS publications, 2011. (**c**) Comparison of field emission properties of FLG grown on different substrates. Reproduced from [73], with the permission of John Wiley and Sons, 2012. (**d**) Dependence of substrate on 2D/1D hybrid field emitter. Reproduced from [60], with the permission of John Wiley and Sons, 2018.

**Figure 7 nanomaterials-13-02437-f007:**
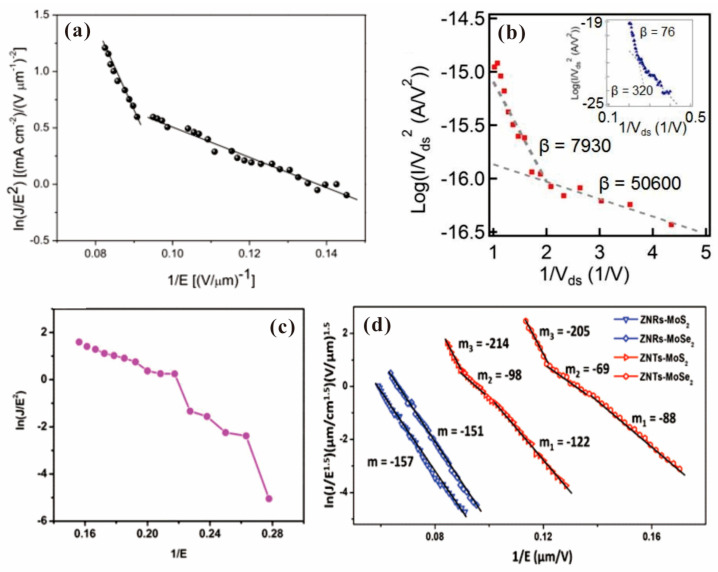
F-N plot from 2D material film. (**a**) Graphene/ZnO nanowire. Reproduced from [57], with the permission of AIP publishing, 2012. (**b**) SnSe. Reproduced from [69], with the permission of John Wiley and Sons, 2019. (**c**) WS_2_. Reproduced from [68], with the permission of Springer Nature, 2013. (**d**) MoS_2_ or MoSe_2_ on ZnO nanostructure. Reproduced from [60], with the permission of John Wiley and Sons, 2018.

**Figure 8 nanomaterials-13-02437-f008:**
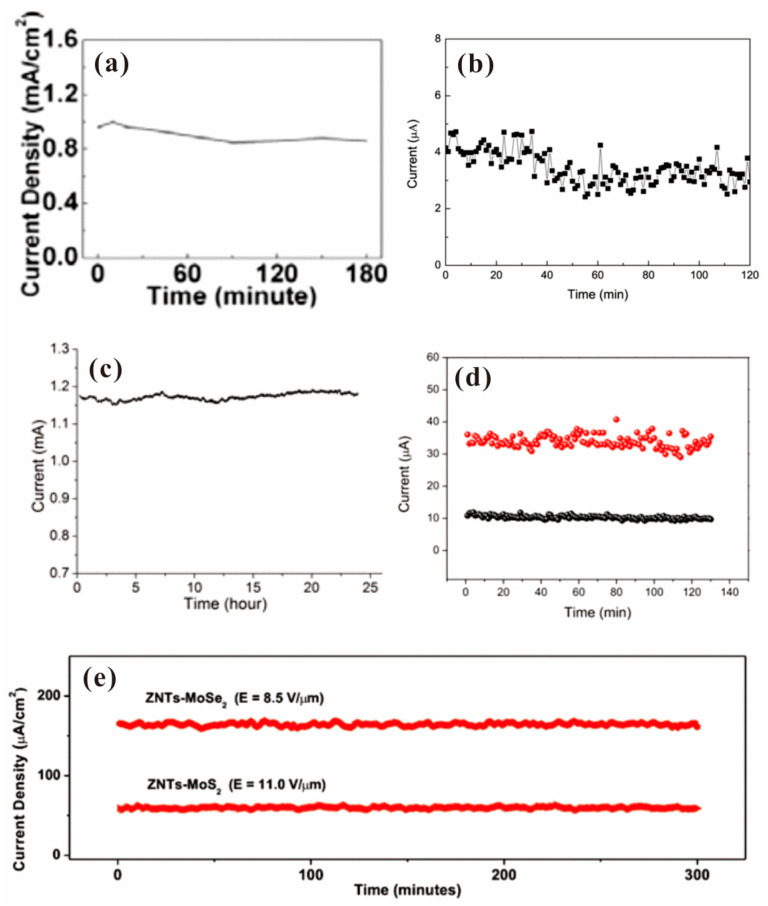
Field emission stability of 2D material film. (**a**) Screen-printed graphene. Reproduced from [35], with the permission of IOP publishing, 2009. (**b**) Exposed graphene. Reproduced from [39], with the permission of AIP publishing, 2011. (**c**) Vertical-aligned FLG. Reproduced from [41], with the permission of IOP publishing, 2012. (**d**) Graphene/ZnO nanowire hybrid emitter. Reproduced from [57], with the permission of AIP publishing, 2012. (**e**) TMD/ZnO nanotaper hybrid emitter. Reproduced from [60], with the permission of John Wiley and Sons, 2018.

**Figure 9 nanomaterials-13-02437-f009:**
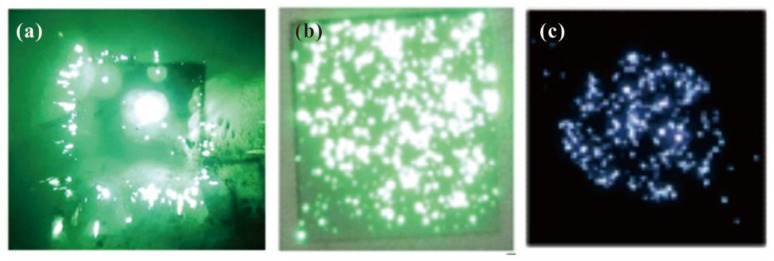
Field emission pattern of several 2D material films. (**a**) Exfoliated BP. Reproduced from [32], with the permission of AIP publishing, 2016. (**b**) Screen-printed FLG. Reproduced from [35], with the permission of IOP publishing, 2009. (**c**) Vertical-aligned FLG. Reproduced from [41], with the permission of IOP publishing, 2012.

**Figure 10 nanomaterials-13-02437-f010:**
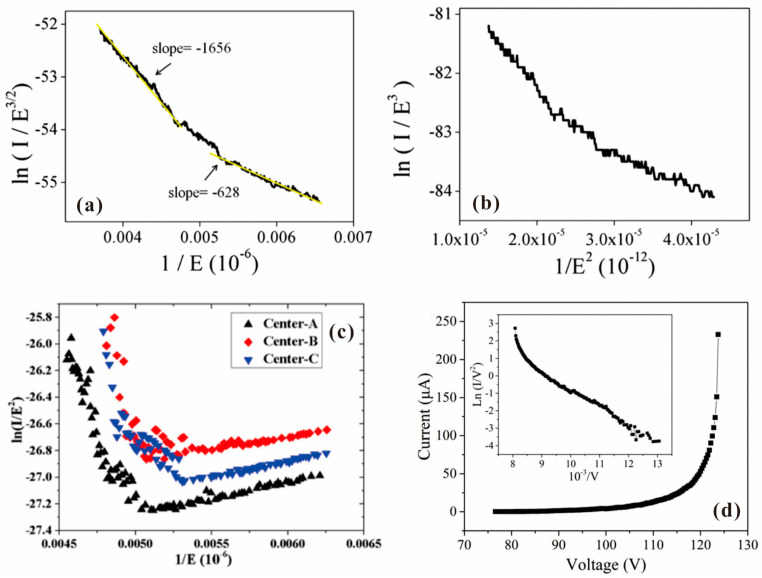
In situ field emission I-V characteristics of graphene. (**a**,**b**) Modified F-N plot of edge of graphene. The units for I and E are pA and MV/m. Reproduced from [74], with the permission of ACS publications, 2010. (**c**) F-N plot of surface emission from suspended graphene. The units for I and E are A and MV/m. Reproduced from [81], with the permission of ACS publications, 2016. (**d**) Field emission I-V curve of FLG. The inset is the corresponding F-N plot. Reproduced from [82], with the permission of John Wiley and Sons, 2021.

**Figure 11 nanomaterials-13-02437-f011:**
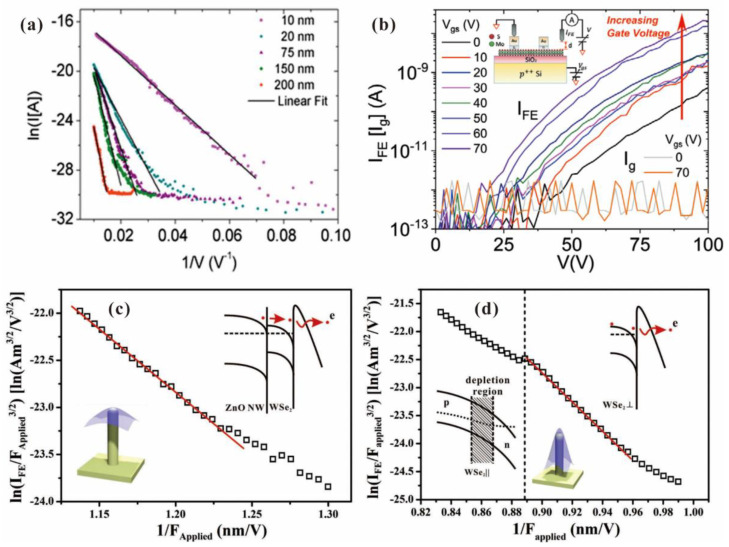
In situ field emission I-V characteristics of other 2D materials. (**a**) MoS_2_ bilayers. Reproduced from [83], with the permission of MDPI, 2018. (**b**) MoS_2_. Reproduced from [84], with the permission of John Wiley and Sons, 2021. (**c**,**d**) WSe_2_ on ZnO nanowire. Reproduced from [85], with the permission of John Wiley and Sons, 2019.

**Figure 12 nanomaterials-13-02437-f012:**
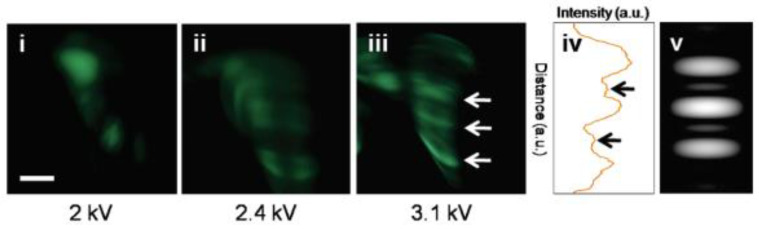
Field emission pattern of RGO under anode voltage of (**i**) 2 kV, (**ii**) 2.4 kV and (**iii**) 3.1 kV. (**iv**) Intensity profile of (**iii**). (**v**) Simulated field emission pattern of three aligned emission sites (scale bar, 5 mm). Reproduced from [22], with the permission of ACS publications, 2011.

**Figure 13 nanomaterials-13-02437-f013:**
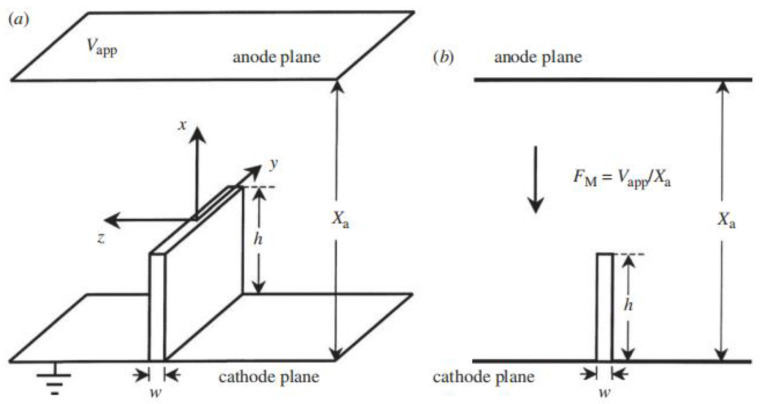
Classical nanowall model. (**a**) Three-dimensional view. (**b**) Projection on z-x plane. Reproduced from [93], with the permission of the Royal Society, 2011.

**Figure 14 nanomaterials-13-02437-f014:**
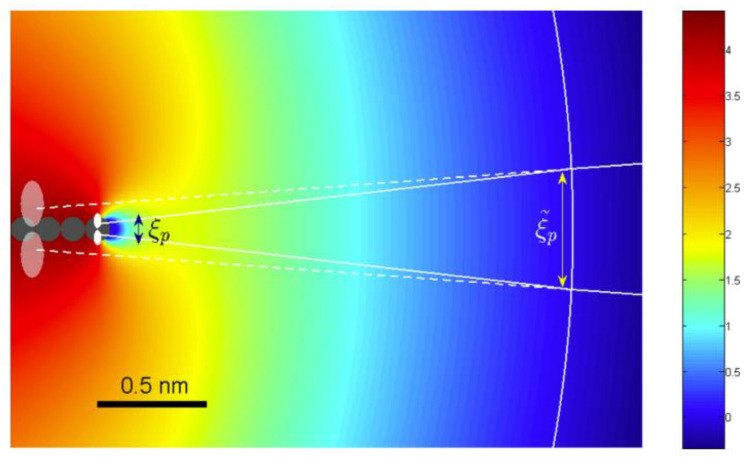
The vacuum energy potential in the z-x plane near the graphene edge emitter. Reproduced from [96], with the permission of AIP publishing, 2016.

**Figure 15 nanomaterials-13-02437-f015:**
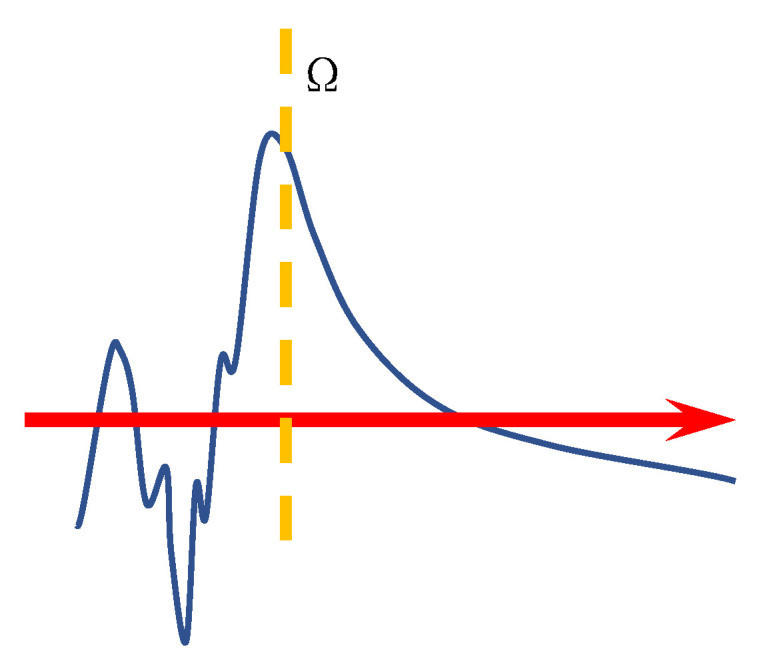
Schematic diagram of the potential distribution near the emission surface.

**Figure 16 nanomaterials-13-02437-f016:**
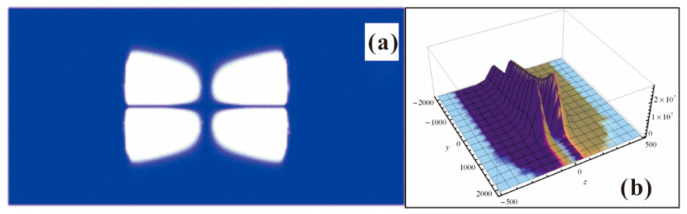
Coherent field emission pattern of graphene. (**a**) Field emission pattern without magnetic field. Reproduced from [95], with the permission of American Physical Society, 2012. (**b**) Field emission pattern with magnetic field. Reproduced from [97], with the permission of AIP publishing, 2014.

**Figure 17 nanomaterials-13-02437-f017:**
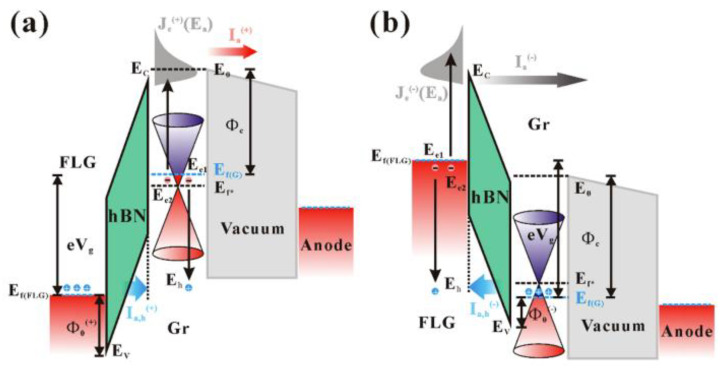
Hot-hole-induced Auger electron emission model for a graphene/hBN/graphene vdW heterostructure. (**a**) The case for a reversed driving voltage. (**b**) The case for a forward driving voltage. Reproduced from [24], with the permission of ACS publications, 2020.

**Table 1 nanomaterials-13-02437-t001:** The state of the art of field emission properties from 2D material films. Note that a general definition for the turn-on field is the field to obtain a current density of 10 μA/cm^2^. In some works, they use a different value of current density in the definition, which has also been listed in the Table.

Materials	Emitter Type	Fabrication Method	Turn-On Field (Current Density)	β	Other Field Emission Behavior
Graphene-based composite thin film	Vertical Edge emitter	Spin coating method	~4 V/μm (10 nA/cm^2^)	1200	Nonlinear F-N plot with current saturation in the high field region was observed [33].
Graphene	Vertical Edge emitter	Electrophoretic method	2.3 V/μm	3700	The I-V curve followed typical F-N tunneling law [34].
	Vertical Edge emitter	Electrophoretic method	4.8 V/μm (0.1 mA/cm^2^)	---	The I-V curves exhibited hysteretic behavior and followed typical F-N tunneling law [75].
	Vertical Edge emitter	Screen printing	1.5 V/μm (1 μA/cm^2^)	4359	The I-V curve followed typical F-N tunneling law [35].
	Vertical Edge emitter	Microfabrication technique	7.2 V/μm (100 nA/cm^2^)	---	The I-V curve followed typical F-N tunneling law [39].
	Lateral Edge emitter	Microfabrication technique	---	~68	The I-V curve followed Child–Langmuir law at low voltage and F-N law at high voltage [50].
	Lateral Edge emitter	Mechanical cutting method	---	---	A strong hysteresis in current–voltage characteristics and a step-like increase in the emission current during voltage ramp up were observed [53].
	Vertical Edge emitter	Adhesive tape treatment	0.73 V/μm	3809	The I-V curve followed typical F-N tunneling law [31].
FLG	Vertical Edge emitter	A solution-based method	7.5 V/μm (0.15 mA/cm^2^)	~1250	The I-V curve followed typical F-N tunneling law [76].
Vertical-aligned FLG	Vertical Edge emitter	MPECVD	~1 V/μm	~7500	The I-V curve followed typical F-N tunneling law [40].
			1.8 V/μm	6795	The I-V curve followed typical F-N tunneling law [41].
	Vertical Edge emitter	PECVD	~5 V/μm	~1750	The I-V curve followed typical F-N tunneling law [42].
	Vertical Edge emitter	IPECVD	---	---	The I-V curve followed typical F-N tunneling law [44].
Vertical-aligned FLG	Vertical Edge emitter	MPECVD and N_2_ plasma treatment	1.94 V/μm	815 (low field) 4710 (high field)	Nonlinear FN plot with two linear sections in the high and low field regions was observed [43].
Vertical-aligned N-doped FLG			1 V/μm	3120 (low field) 17,350 (high field)	
N-doped graphene	Vertical Edge emitter	Arc discharge method	0.6 V/μm	25,849 (low field) 49,690 (high field)	Nonlinear FN plot with two linear sections in the high and low field regions was observed [71].
B-doped graphene			0.8 V/μm	11,879 (low field) 12,067 (high field)	
Graphene			0.7 V/μm	15,740 (low field) 24,058 (high field)	
FLG on CNT	Vertical Edge emitter	Sputtering method	0.956 V/μm	4398	The I-V curve followed typical F-N tunneling law [72,77].
			0.98 V/μm	3980	
FLG on Si microspike array	Vertical Edge emitter	A solution-based method	2.3 V/μm	780–7300	Nonlinear FN plot with current saturation in the high field region was observed [73].
Graphene thin film	Vertical Edge emitter	Mechanical cutting method	---	---	Non-linear FN plot with downward bending in high field region was observed. Three discrete field emission energy peaks existed at low current and they became a single broad spectra at high current [54].
Graphene/hBN/Si	Surface emitter	Transfer method	---	---	The I-V curve followed typical F-N tunneling law [61].
Graphene on ZnO nanowire array	Surface emitter		5.4 V/μm (1 μA/cm^2^)	1100 (high field)	Nonlinear F-N plot with upward bending in the high field region was observed [57].
Graphene on Si tip	Surface emitter		6 V/μm (---)	1000	The I-V curve followed typical F-N tunneling law [58].
GO on Ni nanotip array	Surface emitter		0.5 V/μm (6.7 μA/cm^2^)	---	Upward bending in high field region of the modified FN plot was observed [59].
Graphene on Ni tip	Surface emitter	CVD method	---	---	The I-V curve followed typical F-N tunneling law. A brightness of 1.46 × 10^9^ Am^−2^sr^−1^V^−1^ and energy spread of 0.246–0.42 eV were obtained [6].
WS_2_-RGO nanocomposite	Vertical Edge emitter	Hydrothermal method	2 V/μm (1 μA/cm^2^)	2978	Nonlinear FN plot with current saturation in high field region was observed [68].
WS_2_			3.5 V/μm (1 μA/cm^2^)	2468	
Few-layer MoS_2_	Vertical Edge emitter	Solution-based method	3.5 V/μm	1138	The I-V curve followed typical F-N tunneling law [36].
Few-layer MoS_2_	Vertical Edge emitter	Solution-based method	1 V/μm (1 μA/cm^2^)	9880	The I-V curve followed typical F-N tunneling law [37].
Vertical-aligned MoS_2_	Vertical edge emitter	CVD method	4.5 V/μm	~1064	The I-V curve followed typical F-N tunneling law [78].
Vertical-aligned MoS_2_	Vertical Edge emitter	CVD and transfer method	3.1 V/μm	856	The I-V curve followed typical F-N tunneling law [79].
Vertical-aligned MoS_2_	Vertical Edge emitter	CVD method	~2.46 V/μm	6240	The I-V curve followed typical F-N tunneling law [46].
Vertical-aligned MoSe_2_ on carbon cloth	Vertical Edge emitter	Hydrothermal method	2.4–3.68 V/μm	---	The I-V curve followed typical F-N tunneling law [47].
MoS_2_	Surface emitter	A PMMA-assisted transfer method	9.1 V/μm	---	Three regions can be identified in the modified F-N plot [60].
MoSe_2_			7.0 V/μm	---	
Bi_2_Se_3_ nanosheets	Vertical Edge emitter	Solution-based method	2.3 V/μm	6860	The I-V curve followed typical F-N tunneling law [38].
Bi_2_Se_3_-RGO nanocomposite	Vertical Edge emitter	Hydrothermal method	6 V/μm (1 mA/cm^2^)	---	The maximum field emission current density is 1 mA/cm^2^ when the field is 6 V/μm [80].
Vertical-aligned few-layer ReS_2_	Vertical Edge emitter	Sputtering method	0.8 V/μm (0.6 mA/cm^2^)	~3.3 ×10^5^	The I-V curve followed typical F-N tunneling law [48].
Vertical aligned SnS_2_	Vertical Edge emitter	A biomolecule assisted method	6.9 V/μm	---	The I-V curve followed typical F-N tunneling law [45].
SnSe nanoflowers	Vertical Edge emitter	Solution-based method	---	50,600 (low field) 7930 (high field)	Nonlinear F-N plot with upward bending in the high field region was observed [69].
SnSe single crystal	Surface emitter	Mechanical exfoliation	---	320 (low field) 76 (high field)	
Few-layer BP	Vertical Edge emitter	Mechanical exfoliation	~5.1 V/μm (~1 μA/cm^2^)	1164	The I-V curve followed typical F-N tunneling law [32].
Vertical-aligned CuSe nanosheets	Vertical Edge emitter	An electrochemical method	1.4 V/μm	3545	Nonlinear F-N plot with upward bending in the high field region was observed [49].

**Table 2 nanomaterials-13-02437-t002:** Results of in situ field emission measurement on 2D materials.

Materials	Emission Mode	Remarkable Result
Graphene	edge emission	Field emission I-E characteristics obey ln(I/E^3/2^) versus 1/E^1^ at high field and ln(I/E^3^) versus 1/E^2^ at low field [74].
	surface emission	Field emission current cannot be measured [74].
Graphene	surface emission	A transition process from space charge flow at low bias to the FN theory at high current emission regime was observed [81].
Graphene	surface emission	Field emission current up to 1 μA can be measured at applied field of a few hundred volts per micrometer, which can be well described by the FN model [86].
FLG		
FLG nanosheets	edge emission	A turn-on field of 0.07 V/nm for a field emission current of 1 pA can be obtained [87].
FLG	edge emission	Single FLG with graphite interlayer helps dissipate the joule heat, which can carry a maximum current up to 233 μA [82].
RGO	edge emission	Field emission interference pattern from emission sites separated by a few nanometers is observed, suggesting that the emitted electrons are coherent [22].
WSe_2_	surface emission	In 2D/1D hybrid structure, a “tip contact” structure is favored for achieving lower turn-on field and higher stability due to the hotter injecting electron from 1D nanostructure substrate [85].
WSe_2_	surface emission	The first vacuum transistor based on WSe_2_ monolayer was demonstrated. A turn-on field of ~100 V/μm and good stability were obtained [88].
MoS_2_ bilayers	surface emission	Field emission current is obtained under a field of ~200 V/mm. The result obeys a modified FN model that ln(I) ∝ 1/E [83].
MoS_2_ nanoflower	edge emission	The turn-on field decreases with the cathode–anode distance, which is inconsistent with the FN model [89].
MoS_2_	edge emission	Field emission current can be modulated by using a back gate structure to lower the electron affinity of MoS_2_, which features a new transistor based on field emission [84,90].
MoS_2_ bilayer	surface emission	Their field emission characteristics follow a 2D modified FN model [91].
WSe_2_ monolayer		

## Data Availability

No new data were created or analyzed in this study. Data sharing is not applicable to this article.

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
