# Peer review of "Cold Cathodes with Two-Dimensional van der Waals Materials"

_nanomaterials, 2023, doi:10.3390/nano13172437_

Round 1
Reviewer 1 Report
The manuscript "Cold cathodes with two-dimensional van der Waals materials" provides an overview of cold cathodes with two-dimensional van der Waals materials. The fabrication process of some emitter structures for field emission applications, the state-of-the-art of their field emission properties, and existing models are described. A number of deficiencies have been identified in the manuscript, the correction of which will greatly improve the quality of the published material:
1. Section 2.1.1. does not provide a more detailed description of the vertical emitter, nor does it explicitly identify the benefits derived from this structure.
2. Section 2.1.2. does not provide a more general description of the lateral structure.
3. The general logic of the article seems somewhat inconvenient. It would be more practical to provide a description of each structure considered in the article, its fabrication process, and its field emission properties all at once.
4. Tables 1 and 2 present the characteristics of various emitters of different types, but no conclusions are drawn from the tables. It is necessary to add conclusions about the efficiency of emitters based on emission indicators such as emission current density and turn-on field.
5. In recent years, simple methods for the fabrication and laser structuring of field emitters based on thin films of carbon tubes and graphene have been developed [doi:10.1016/j.diamond.2019.04.035; doi:10.3390/nano12162812]. Adding references to these works would significantly improve the quality of the manuscript.
6. Section 5 does not reflect the advantages of vertical and surface emitters.
7. It is necessary to add the decoding of the abbreviations CVD, PMMA.
8. Figure 5(b) is missing a dimension ruler.
9. Figures 10(c and b) need an explanation of what 38 is in the legend.
10. Figures 10 and 11 are of insufficient quality.
Author Response
Answers to the reviewer’s comments and list of revisions
The reviewers have given important comments and questions on this manuscript. The answers to them are given as follows.
The comments and questions of the reviewers are given in italics. Our answers are given following each point.
Answers to the comments of Reviewer #1
The manuscript "Cold cathodes with two-dimensional van der Waals materials" provides an overview of cold cathodes with two-dimensional van der Waals materials. The fabrication process of some emitter structures for field emission applications, the state-of-the-art of their field emission properties, and existing models are described. A number of deficiencies have been identified in the manuscript, the correction of which will greatly improve the quality of the published material:
1, Section 2.1.1. does not provide a more detailed description of the vertical emitter, nor does it explicitly identify the benefits derived from this structure.
- Thanks for pointing out this problem.
- A vertical emitter is defined as that edge structure is vertical to the cathode substrate.
- Compared to the lateral emitter, vertical aligned emitter is more suitable in the application of flat panel electron source since it can be fabricated with multi emitters on a flat substrate.
- The corresponding description has been added. (Lines 97-98 in the revised manuscript)
2, Section 2.1.2. does not provide a more general description of the lateral structure.
- A lateral emitter is defined as that the edge structure is parallel with the cathode substrate.
- The corresponding description has been added. (Lines 97-98 in the revised manuscript)
3, The general logic of the article seems somewhat inconvenient. It would be more practical to provide a description of each structure considered in the article, its fabrication process, and its field emission properties all at once.
- Thanks for the suggestion.
- In fact, we have thought about the logic suggested by the reviewer. However, it seems inconvenient to make the comparison on the field emission properties (such as F-N plot, stability and uniformity) of different structure under that logic. Besides, since the theoretical model for 2D field emitter is an important part of our review, and some of the theoretical prediction has not been observed experimentally, it is difficult to put them under the part of a certian structure. Based on the above considerations, we divided the review into two parts: experiment and theory. In the part of experiment, it is divided into emitter structure and field emission properties.
4, Tables 1 and 2 present the characteristics of various emitters of different types, but no conclusions are drawn from the tables. It is necessary to add conclusions about the efficiency of emitters based on emission indicators such as emission current density and turn-on field.
- Thanks for the suggestion.
- From Table 1, it is clearly seen that the turn-on field for vertical edge emitter is much smaller than that for surface emitter, which is attributed to its larger field enhancement factor. This indicates a higher emission efficiency for the vertical edge emitter.
- From Table 2, it is seen that different from the film measurement, most of the results in the in-situ measurement don’t follow the classical F-N law, which reflect the intrinsic field emission properties of 2D emitter.
- The corresponding description has been added. (Lines 345-348 and 418-420 in the revised manuscript)
5, In recent years, simple methods for the fabrication and laser structuring of field emitters based on thin films of carbon tubes and graphene have been developed [doi:10.1016/j.diamond.2019.04.035; doi:10.3390/nano12162812]. Adding references to these works would significantly improve the quality of the manuscript.
- The references have been added. (Reference 105 and 106 in the revised manuscript)
6, Section 5 does not reflect the advantages of vertical and surface emitters.
- Based on the review, it is seen that a vertical edge emitter has a high emission efficiency while a surface emitter has a high stability. Depending on the requirement of applications, one may choose different type of emitter as the cathode. Further optimization on their field emission properties may also be per-formed by using post treatment method such as laser structuring method.
- The corresponding discussion has been added. (Lines 718-723 in the revised manuscript)
7, It is necessary to add the decoding of the abbreviations CVD, PMMA.
- The corresponding decoding of the abbreviations CVD and PMMA have been added. (Lines 134 and 193 in the revised manuscript)
8, Figure 5(b) is missing a dimension ruler.
- Thanks for pointing out this problem.
- The description for the scale bar has been added. (Line 245 in the revised manuscript)
9, Figures 10(c and b) need an explanation of what 38 is in the legend.
- “38” in Figs. 10(a) and 10(b) represent the measurement time of the result. Since it may make confusion for the readers, “38” in the figure has been eliminated in the revised manuscript.
10, Figures 10 and 11 are of insufficient quality.
- The quality of figures 10 and 11 have been improved in the revised manuscript.
Reviewer 2 Report
This paper is a very interesting and quite comprehensive review of electron field emission by 2D van der Waals materials, with experimental and theoretical details. In my opinion, it certainly deserves publication, possibly with a few minor improvements and some English editing, as detailed below.
Concerning section 2, I note that there are no values on the scale bars of Figs. 1d) and 5b). Even if they are not present in the original figures, they could maybe be inferred from the legends or the main text in the corresponding articles and added in the legends of Figs. 1 and 5 of the present paper.
Concerning section 3,
- at line 267 is the turn-on field really 0.6 V/mm or 0.6 V/µm?
- At line 274 the maximum current intensity of 7 A/cm^2 could perhaps be compared to corresponding values for other kinds of field emission flat displays such as e.g. CNT arrays. Also at line 277, it would be interesting to have the area from which the 35 µA were obtained.
- Furthermore, at line 305, the upward bending in the high field region of Fowler-Nordheim (F-N) plot could possibly be compared to similar features for 1D emitters.
- I appreciated the discussion of stability starting at line 319. However, it made me thing that the question of uniformity mentioned at lines 333 and 334 could also be discussed in more details, since it is a currently important limitation for applications.
- In Table 1, it is not clear to me what the numbers quoted between parentheses for current density really are and why the I-V curves or F-N plots (of e.g. Figs 6 or 7) could not be used to give values for more materials, if it is a maximum value.
- On Fig. 10, units for subfigures a), b) and c) should be given in the legend.
Concerning section 4,
- It seems to me that there are more English problems in this paragraph that in the previous ones (conners => corners, buck => bulk, inversion between ‘ed’ and ‘ing’ at the end of verbs, Greens’ => Green’s, as height as => as high as, phonon assistant => phonon assisted, …). Furthermore, the equation editor should also be used systematically for variable names (t, h, F, H_0…) and small inline equations in the text (with same size of characters as in the text)
- JWKB is not defined the first time it is used (so as SEM at line 138 and TEM at line 352). I understand these are very classic acronyms, but other classic acronyms were defined ;-)
- The definition of gamma at lines 471-472 is not clear to me (possibly just a problem of English wording).
- For the divergence angle at lines 521 and 530, why is there a factor 2 in the first case and not in the second one?
- In equation 8, omega should also be explicitly mentioned on the right hand-side and the (too small!) sign at the right of the reduces Planck constant should be explained.
- At lines 584-585, I would expect more comments about the differences between the results of Luo et al. and other results that do not incorporate optical excitation, phonon scattering, and thermal relaxation.
- Similarly, I would have expected more (possibly quantitative) comparisons between theoretical and experimental results (e.g. for Fig. 16) or concerning explanations of the non standard F-N plots mentioned in Table 1, or some words explaining that there are no similar experimental results (as mentioned in section 5)
- Is there really no ab-initio study of field emission by 2D materials?
Concerning section 5,
- For the fact that edge emission in 2D materials is significantly weaker than for nanotips, one simple reason would be that the field enhancement factor is said to be of the order of sqrt(height/width) in Eq. (1) while, if I remember well, it is of the order of height/width for 1D emitters?
Concerning references, quite of lot of letters that should be upper case in the titles are in lower case in the submitted paper: x-ray => X-ray, cnt=> CNT, cuo=> CuO, zno=> ZnO, van der waals => van der Waals, auger=> Auger.
Some English editing is necessary, particularly (but not only) concerning section 4 (cf. my suggestions above)
Author Response
Answers to the comments of Reviewer #2
This paper is a very interesting and quite comprehensive review of electron field emission by 2D van der Waals materials, with experimental and theoretical details. In my opinion, it certainly deserves publication, possibly with a few minor improvements and some English editing, as detailed below.
1, Concerning section 2, I note that there are no values on the scale bars of Figs. 1d) and 5b). Even if they are not present in the original figures, they could maybe be inferred from the legends or the main text in the corresponding articles and added in the legends of Figs. 1 and 5 of the present paper.
- Thanks for pointing out this problem.
- In fact, the scale bar in Fig. 1d is very small so it is not easy to be found. To make a better presentation, Fig. 1d has been modified. Besides, the description for the scale bar in Fig. 5b has also been added.
2, Concerning section 3, at line 267 is the turn-on field really 0.6 V/mm or 0.6 V/µm?
- Thanks for pointing out this mistake.
- The turn-on field is 0.6 V/µm.
- The corresponding description has been replaced. (Line 269 in the revised manuscript)
3, At line 274 the maximum current intensity of 7 A/cm^2 could perhaps be compared to corresponding values for other kinds of field emission flat displays such as e.g. CNT arrays. Also at line 277, it would be interesting to have the area from which the 35 µA were obtained.
- We believe the reviewer may have made a mistake. The maximum current density from vertical aligned FLG is 7 mA/cm2, which is smaller by several orders of magnitude than the reported value from CNT arrays (> 1000 mA/cm2) [1].
- The area from which the 35 µA at line 277 is estimated as 0.00449 µm2 in the reference.
- The corresponding value for the area has been added. (Lines 279-280 in the revised manuscript)
Reference
[1] Fairchild, S. B.; Zhang, P.; Park, J.; Back, T. C.; Marincel, D.; Huang, Z.; Pasquali, M. Carbon nanotube fiber field emission array cathodes. IEEE Transactions on Plasma Science 2019, 47, 2032-2038.
4, Furthermore, at line 305, the upward bending in the high field region of Fowler-Nordheim (F-N) plot could possibly be compared to similar features for 1D emitters.
- Thanks for the suggestion.
- In fact, not only the upward bending but also the downward bending in the F-N plot is similar with the case in 1D emitter.
- Generally, the underlying mechanism of nonlinear F-N plot can be attributed by current saturation, joule heat, surface state emission and electron confinement which can exist in both cases of 1D and 2D emitter.
- The corresponding discussion has been added. (Lines 304-307 in the revised manuscript)
5, I appreciated the discussion of stability starting at line 319. However, it made me thing that the question of uniformity mentioned at lines 333 and 334 could also be discussed in more details, since it is a currently important limitation for applications.
- For further improving the uniformity, one needs to fabricate the 2D emitter with not only a uniform spatial distribution but also a uniform surface work function as well as a uniform resistance. Using surface emitter such as 2D/1D hybrid emitter may be a solution due to its net-like structure. But related result has not been reported yet.
- The corresponding discussion has been added. (Lines 339-343 in the revised manuscript)
6, In Table 1, it is not clear to me what the numbers quoted between parentheses for current density really are and why the I-V curves or F-N plots (of e.g. Figs 6 or 7) could not be used to give values for more materials, if it is a maximum value.
- Generally, the definition for the turn-on field is the field to obtain a current density of 10 µA/cm2. However, in some works, they use a different value of current density in the definition. The current density behind the turn-on field in Table 1 is the corresponding value for the definition of turn-on field in that work.
- We are not sure about whether the reviewer’s question is why not use the I-V curves to estimate the turn-on field or field enhancement factor in Table 1, since some works haven’t provided the data. For this question, it is because only the I-V curve is not enough to extract the turn-on field if the emission area and the anode-cathode distance are unknown. In some case that the F-N plot is nonlinear, it is also difficult to extract the field enhancement factor since the F-N slope varies with the field.
- The corresponding description has been added in the caption of Table 1 to avoid confusion. (Lines 360-362 in the revised manuscript)
7, On Fig. 10, units for subfigures a), b) and c) should be given in the legend.
- The units for I and E in Fig. 10(a) and 10(b) are pA and MV/m, while the units for I and E in Fig. 10(c) are A and MV/m. Since the original figures haven’t contained information about the unit, they have been added in the caption of the figure. (Lines 384-386 in the revised manuscript)
8, Concerning section 4, It seems to me that there are more English problems in this paragraph that in the previous ones (conners => corners, buck => bulk, inversion between ‘ed’ and ‘ing’ at the end of verbs, Greens’ => Green’s, as height as => as high as, phonon assistant => phonon assisted, …). Furthermore, the equation editor should also be used systematically for variable names (t, h, F, H_0…) and small inline equations in the text (with same size of characters as in the text)
- The English problem and the equations have been corrected. (Lines 458, 465, 466, 527, 533, 534, 536, 548, 564, 574, 579, 580, 588, 649, 666, 670, 671, 673, 684, 685, 691, 698 in the revised manuscript)
9, JWKB is not defined the first time it is used (so as SEM at line 138 and TEM at line 352). I understand these are very classic acronyms, but other classic acronyms were defined ;-)
- The corresponding definitions for JWKB, SEM and TEM have been added. (Lines 489, 139 and 365 in the revised manuscript)
10, The definition of gamma at lines 471-472 is not clear to me (possibly just a problem of English wording).
- The gamma γ is the angle of radius vector with respect to one of the wedge plane.
- This definition is used instead of the original one. (Lines 487-488 in the revised manuscript)
11, For the divergence angle at lines 521 and 530, why is there a factor 2 in the first case and not in the second one?
- Thanks for pointing out this mistake. The factor 2 is missed in the second one. In present paper, the divergence angle is defined as the angle of two asymptotic profile lines of the beam in the z-x plane, so it is 2 times of the angle of one of the asymptotic profile line to the x-axis. The latter is defined as the divergence angle in [96].
- We have corrected this mistake. (Line 550 in the revised manuscript)
12, In equation 8, omega should also be explicitly mentioned on the right hand-side and the (too small!) sign at the right of the reduces Planck constant should be explained.
- Thanks for the suggestions. We have changed the form of the equation to make it clearer. More description about the equation has been given. (Lines 590-594 in the revised manuscript)
13, At lines 584-585, I would expect more comments about the differences between the results of Luo et al. and other results that do not incorporate optical excitation, phonon scattering, and thermal relaxation.
- In the quasi-equilibrium approximation of field emission without optical excitation, the occupation probability of electrons follows the Dirac-Fermi distribution. Under a continuous laser, the steady distribution of electrons depends on the laser and the phonon scattering. Therefore, the emission pattern should be changed in principle.
- Comments have been added. (Lines 605, 612-621 in the revised manuscript)
14, Similarly, I would have expected more (possibly quantitative) comparisons between theoretical and experimental results (e.g. for Fig. 16) or concerning explanations of the non standard F-N plots mentioned in Table 1, or some words explaining that there are no similar experimental results (as mentioned in section 5)
- Some comments about the comparison between theoretic and experimental results have been added. (Lines 494-495 and 622-634 in the revised manuscript)
15, Is there really no ab-initio study of field emission by 2D materials?
- Yes, there are some ab-initio studies on field emission from 2D materials. Ref. [55] is one of them. We have also added some comments and provided a new reference. (Lines 629-634 in the revised manuscript)
16, Concerning section 5, For the fact that edge emission in 2D materials is significantly weaker than for nanotips, one simple reason would be that the field enhancement factor is said to be of the order of sqrt(height/width) in Eq. (1) while, if I remember well, it is of the order of height/width for 1D emitters?
- We have added this comment in (i) of the section 5. (Lines 730-732 in the revised manuscript)
17, Concerning references, quite of lot of letters that should be upper case in the titles are in lower case in the submitted paper: x-ray => X-ray, cnt=> CNT, cuo=> CuO, zno=> ZnO, van der waals => van der Waals, auger=> Auger.
- The corresponding letters have been replaced by the upper case.
18, Some English editing is necessary, particularly (but not only) concerning section 4 (cf. my suggestions above)
- Thanks for the suggestion.
- We have checked the English carefully on the revised manuscript.

Round 2
Reviewer 1 Report
I think the manuscript may be published.